# Discrimination of biofilm-producing *Stenotrophomonas maltophilia* clinical strains by matrix-assisted laser desorption ionization–time of flight

Edeer Montoya-Hinojosa[1], Paola Bocanegra-Ibarias[2], Elvira Garza-González[2], Óscar Manuel Alonso-Ambriz[1], Gabriela Aimee Salazar-Mata[1], Licet Villarreal-Treviño[1], Eduardo Pérez-Alba[2], Adrián Camacho-Ortiz[2], Rayo Morfín-Otero[3], Eduardo Rodríguez-Noriega[3], Samantha Flores-Treviño[2]*

1 Facultad de Ciencias Biológicas, Universidad Autónoma de Nuevo León, NL, México, 2 Hospital Universitario Dr. José Eleuterio González, Universidad Autónoma de Nuevo León, NL, México, 3 Hospital Civil de Guadalajara Fray Antonio Alcalde e Instituto de Patología Infecciosa y Experimental, Centro Universitario de Ciencias de la Salud, Universidad de Guadalajara, JAL, México

* samflorest@gmail.com

## Abstract

*Stenotrophomonas maltophilia* is a Gram-negative drug-resistant pathogen responsible for healthcare-associated infections. The aim was to search for biomarker peaks that could rapidly detect biofilm production in *S. maltophilia* clinical isolates obtained from two tertiary care hospitals in Mexico. Isolates were screened for the presence of biofilm-associated genes, in which the *fsnR* gene was associated with biofilm production ($p = 0.047$), whereas the *rmlA⁺* genotype was associated with the *rpfF⁻* genotype ($p = 0.017$). Matrix-assisted laser desorption ionization–time of flight (MALDI-TOF) mass spectra comparison yielded three potential biomarker peaks (4661, 6074, and 6102 *m/z*) of biofilm-producing *rmlA⁺* and *rpfF⁻* genotypes with >90% sensitivity ($p<0.001$). MALDI-TOF MS analyses showed a correlation between the relative abundance of 50S ribosomal proteins (L30 and L33) and the presence of the *fnsR*, *rmlA* and *rpfF*-2 genes, suggested to play a role in biofilm formation. Isolates obtained in the intensive care unit showed low clonality, suggesting no transmission within the hospital ward. The detection of biomarkers peaks by MALDI-TOF MS could potentially be used to early recognize and discriminate biofilm-producing *S. maltophilia* strains and aid in establishing appropriate antibiotic therapy.

## Introduction

*Stenotrophomonas maltophilia* is a non-fermenting Gram-negative bacillus, which causes healthcare-associated infections and exhibits increasing drug resistance rates [1, 2]. Infections caused by *S. maltophilia* mostly occur in the lower respiratory tract (up to 55.8%), though bloodstream (33.8%), skin and skin structure (7.8%), urinary tract (1.2%), and intra-

**Data Availability Statement:** All relevant data are within the paper and its Supporting Information files.

**Funding:** This work was supported by the Fondo Sectorial de Investigación para la Educación of the Consejo Nacional de Ciencia y Tecnología (CONACyT) [grant number 284041], received by SFT. The funders had no role in study design, data collection and analysis, decision to publish, or preparation of the manuscript.

**Competing interests:** The authors have declared that no competing interests exist.

abdominal infections (1.0%) are also frequent [3, 4]. Among the risk factors of acquisition of *S. maltophilia* infections are those which compromise the immune system, such as the presence of an underlying malignancy or organ transplantation, prolonged hospitalization and/or intensive care unit (ICU) admission, cystic fibrosis, the presence of medical indwelling devices, and prior antibiotic intake [1, 3].

*S. maltophilia* produces high levels of biofilm in most infections, which helps the bacteria to facilitate colonization, evade the host immune response and resist the effect of the antibiotics [5]. Infections with biofilm-producing strains are difficult to treat and eradicate, as they rarely respond to conventional antibiotics. In addition to bacterial adhered cells, the biofilm is mostly composed of extracellular matrix material, also known as extracellular polymeric substances (EPS). The components of this biofilm matrix, which includes a mixture of polysaccharides, proteins, nucleic acids, and lipids [6] that varies among bacterial species, provide structure and protect the cells inside the biofilm against external stress conditions, such as the host immune system or the effect of antibiotics [5, 7].

Matrix-assisted laser desorption ionization–time of flight mass spectrometry (MALDI-TOF MS) is becoming increasingly important for bacterial identification instead of conventional phenotypic methods. In addition, this method allows the discrimination of biofilm-producing strains [8], prompting an early and effective therapy.

Epidemiological surveillance studies of *S. maltophilia* infections within the hospital setting in several regions worldwide indicate low infection transmission [9]. As high genetic diversity is generally exhibited by this pathogen [10], several sources of infection are considered, including those from the environment [11]. As ICU-associated *S. maltophilia* infections seem to behave differently than those associated to the community [12], we sought to investigate this matter in our hospital setting. Therefore, the aim of this study was to characterize biofilm production and genetic diversity of *S. maltophilia* clinical isolates obtained over a seven-year period from the ICU of two tertiary care hospitals in Mexico. We also searched for protein biomarker peaks of specific biofilm-associated genotypes using MALDI-TOF MS.

## Material and methods

### Study design

The strains were obtained from 2011 to 2017 in two third-level hospitals in Mexico. The Hospital Universitario Dr. José Eleuterio González is a 500-bed teaching hospital located in Monterrey (Nuevo Leon), the third largest city in Northeastern Mexico. It receives referrals from neighboring state hospitals, and it has approximately 23,000 admissions per year and 200,000 emergency room visits yearly. The Hospital Civil de Guadalajara Fray Antonio Alcalde is a 1,000-bed teaching hospital located in Guadalajara (Jalisco), the second largest city in western Mexico. It receives referrals from neighboring state hospitals, and it has approximately 35,000 admissions per year and 450,000 emergency room visits yearly. Only one isolate per patient was included. Clinical and demographic data from each patient was obtained.

### Ethics statement

This study was performed with approval from the Ethics Committee of the Hospital Universitario Dr. José Eleuterio González (approval number GA20-00001) and the Hospital Civil de Guadalajara Fray Antonio Alcalde (approval number 011/14). The committee approved a request to waive patients' written consent because the clinical information was anonymized, and the bacterial isolates were collected in a previous study where subjects were informed about the collection and potential future use of biospecimens (approval number GA15-005).

## Culture and identification of *S. maltophilia* isolates

Bacteria were grown on plates containing trypticase soy agar with 5% sheep blood and incubated overnight at 35°C. All *S. maltophilia* isolates were stored at −70°C until further use. *S. maltophilia* ATCC 13637 was used as the reference strain.

Cultures were identified by matrix-assisted laser desorption ionization–time of flight mass spectrometry (MALDI-TOF MS, Microflex LT system, Bruker Daltonics, Bremen, Germany) according to the manufacturer's recommendations. One colony from an overnight culture grown on blood agar at 37°C under aerobic conditions was applied with a sterile wooden toothpick on a 96-spot stainless steel target plate (Bruker Daltonics, Bremen, Germany). After drying, 1 μL 70% formic acid was added and air-dried prior to adding 1 μL of matrix solution (10 mg/mL α-cyano-4-hydroxycinnamic acid in 50% acetonitrile and 2.5% trifluoroacetic acid [Sigma-Aldrich, Toluca, Mexico]). The spots were air-dried, and the target plate was introduced into the equipment. The MALDI Biotyper 3.0 software was used to match the spectra profile with the database. Classification was performed according to the manufacturer's recommended score identification criteria, in which a score within 2.0–3.0 range indicates reliable species-level identification.

## Biofilm production and EPS composition determination

Biofilm formation was determined ($n$ = 120) using the crystal violet staining method, in which 200 μL of a 1.0 McFarland standard suspension 1:100 dilution from overnight cultures in tryptic soy broth was inoculated into a 96-well flat-bottom polystyrene plate. After 24 h incubation at 37°C, the optical density at 595nm ($OD_{595nm}$) of planktonic cells was measured in a spectrophotometer (iMark, Bio-Rad Laboratories Inc, CA, USA). After washing with PBS (pH 7.3) three times to remove planktonic cells, the biofilms were staining with 200 μL 0.5% crystal violet for 5 min. The wells were washed again, added 150 μL 33% glacial acetic acid for 15 min, and measured at $OD_{595nm}$. The biofilm index (BI) was obtained using the $OD_{595nm}$ of the planktonic cells/ $OD_{595nm}$ of the biofilm. The strains were classified as weak (BI ≤ 0.5), moderate (BI>0.5≤1), and strong (BI >1) biofilm producers.

Biofilm-matrix detachment assays were performed to determine EPS composition in selected biofilm-producing isolates ($n$ = 120). Sodium metaperiodate ($NaIO_4$, 40 mM), proteinase K (0.1 mg/mL), and DNase I (0.5 mg/mL in 5 mM $MgCl_2$) were used to degrade β-1,6-linked polysaccharides, proteins, and extracellular DNA, respectively [13]. Briefly, 24 h-old biofilms in 96-well microtiter plates were first rinsed three times with PBS 1X and incubated with one of the three aforementioned degrading component (treated wells) or tryptic soy broth solutions (untreated wells) for 24 h at 37°C. Afterward, the solutions were discarded, the biofilms were stained with crystal violet 0.5% as described above, and $OD_{595}$ was measured. The percentage of biofilm detachment was calculated based on the average difference between the treated and the untreated wells. Biofilm detachment values above 75% of either polysaccharides, proteins, or extracellular DNA were considered as major components.

## Biofilm-associated genes detection

Isolates were screened for the presence of *fsnR*, *rmlA*, *rpfF-2*, and *xanB* genes using PCR conditions previously described [14, 15]. Primers were used for *fsnR* (forward: 5′-AGATCTCCGCCAA GATGCTG-3′ and reverse: 5′-CCAGTGACTCCATCATGCGT-3′), *rmlA* (forward: 5′-CGGAA AAGCAGAACATCG-3′ and reverse: 5′-GCAACTTGGTTTCAATCACTT-3′), *rpfF-2* variant (forward: 5′-CACGACAGTACAGGGGACC-3′ and reverse: 5′-GGCAGGAATGCGTTGG-3′), and *xanB* (forward: 5′-ATGGTCGGCCTGGAAAATGT-3′ and reverse: 5′-TTCTTCAGGCGAT GGGTGAC-3′) genes. Briefly, the reaction mixtures contained 1X PCR buffer, 2 mM $MgCl_2$,

200 nM of each dNTP, 200 nM of each primer, 1 U AmpliTaq polymerase (Bioline, Taunton, MA), and 5 uL of DNA extracted by thermal lysis. PCR was initiated by denaturation for 1 min at 94˚C, followed by 30 (35 for *fsnR* gene) cycles of 1 min denaturation at 94˚C, 1 min annealing at 53–57˚C, and 1 min extension at 72˚C, with a final extension of 5 min at 72˚C. Expected *fsnR*, *rmlA*, *rpfF-2*, and *xanB* PCR products were 192 bp, 1222 bp, 1140 bp, and 374 bp, respectively.

## Clonal diversity determination

The genomic DNA from selected isolates (*n* = 50, ICU-obtained) was extracted and digested with *Xba*I restriction enzyme. Pulsed field gel electrophoresis (PFGE) was performed using the CHEF-DR III system (Bio-Rad, CA, United States) in which a 1% agarose gel was run at 14˚ C with an initial time of 0.5 s and final time of 35 s, at 6V for 20 h. The band patterns generated with the Labworks 4.5 software were analyzed with 1% tolerance using SPSS software version 25.0 (IBM Corporation, NY, United States). The similarity coefficients were generated using the Jaccard coefficient.

## Multilocus sequence typing

Three randomly selected strains per each year of the study (*n* = 14) were subjected to multilocus sequence typing (MLST). The sequence types (ST) were obtained using the conditions and the primers for seven housekeeping genes (*atpD*, *gapA*, *guaA*, *mutM*, *nuoD*, *ppsA* and *recA*) suggested in the *Stenotrophomonas maltophilia* MLST website (https://pubmlst.org/smaltophilia/). The products were sequenced on a 3730xl DNA Analyzer (Applied Byosistems, ThermoFisher Scientific, MA, United States) by Macrogen Inc. (Seoul, Korea). The sequences were assembled in the BLAST program (NCBI) [16] and submitted to the database to determine the ST.

## Biomarker peaks determination and identity assignment

The mass spectra obtained from each isolate after MALDI-TOF MS analysis were processed using flexAnalysis software (Bruker Daltonics) in which top-hat baseline subtraction, spectra smoothing (with the Savitzky/Golay algorithm for 10 cycles with 2 *m/z* width), and normalization (the average spectrum of each group was subjected to peak picking, with a signal to noise threshold of 5) were performed, and they were then imported into the ClinProTools V.3.0 software (Bruker Daltonics) for recognition of mass spectra patterns between groups. Peak selection was performed using the P-value T-test/ANOVA sort mode. Group selection was based on the biofilm production and the presence of biofilm-associated genes.

MS spectra were exported to the Biotools 3.2 software (Bruker Daltonics) in which peptide/protein assignment was performed by the Mascot Server (Matrix Science, Boston, USA) and run against the SwissProt database.

## Statistical analyses

The Anderson-Darling, Student t, and Wilcoxon tests included within the ClinProTools software were used in the selection of biomarker peaks. To determine the distribution of the population (≤0.05: not normally distributed, >0.05, normally distributed), the Anderson-Darling test was used. Either the t-test (used for normally distributed data) or the Wilcoxon test (user for not normally distributed data) were used to confirm significant differences between two classes (biofilm-producing versus not biofilm-producing phenotype). For each test, a *p* value ≤0.05 was considered significant and thus the peak was confirmed to be significantly different. Peaks with statistical significance were further analyzed. The peak area and/or intensity of the

spectra were evaluated as well as the coefficient of variation of each peak. ClinProTools software also calculates a Receiver Operating Characteristic (ROC) curve for each peak, which provides an evaluation of the discrimination quality of the peak. Furthermore, only peaks with values over 0.80 for the area under the curve (AUC) were selected. After obtaining statistical significance and evaluating the coefficient of variation and the intensity of each peak, potential biomarker peaks were selected.

The sensitivity, specificity, positive predictive value (PPV), and negative predictive value (NPV) of each peak were also evaluated for potential biomarker peaks.

## Results

### *S. maltophilia* clinical isolates

During the 7-year study period, 120 *S. maltophilia* clinical isolates from Nuevo León (n = 100, 83.3%) and Jalisco (n = 20, 16.7%) were analyzed (S1 Table). More than a third of the patients were in the intensive care unit (n = 47, 38.8%), and none had cystic fibrosis. The majority of the studied subjects were men (n = 66; 55.0%), and the age ranged from 18 to 87 years, with a mean of 46.2 years. The majority of the isolates were from the respiratory tract (n = 73; 60.8%), followed by blood (n = 16; 13.3%), catheter (n = 5; 4.2%), wounds (n = 4; 3.3%), biopsy (n = 4; 3.3%), urine (n = 3; 2.5%), pleural fluid (n = 3; 2.5%), abscesses (n = 2; 1.7%), and cerebrospinal fluid (n = 1; 0.8%).

### Genetic diversity

PFGE analysis showed 49 different patterns and two isolates with identical patterns (clone A), both of which were obtained from respiratory samples in two female patients from the same hospital (Jalisco), within a three-month period of ICU admission. The percentage of similarity of all isolates ranged between 75% to 100%. MLST analysis showed 12 different ST and two isolates belonging to ST186, which also corresponded to clone A (S1 Fig).

### Biofilm production and EPS composition

As shown in Table 1, all *S. maltophilia* isolates presented biofilm production. The biofilm was characterized as weak in 17.5% (21/120) of the isolates; moderate, 44.2% (53/120); and strong,

**Table 1. Phenotypic and genotypic characteristics of biofilm production in *S. maltophilia* isolates.**

| | | | Biofilm-associated genes | | | |
| --- | --- | --- | --- | --- | --- | --- |
| | | | *n* (%) | | | |
| | | n (%) | *fsnR* | *rmlA* | *rpfF-2* | *xanB* |
| **Biofilm production** | Weak | 21 (17.5) | 20 (16.7) | 16 (13.3) | 9 (7.5) | 3 (2.5) |
| | Moderate | 53 (44.2) | 53 (44.2) | 50 (41.7) | 18 (15.0) | 9 (7.5) |
| | Strong | 46 (38.3) | 41 (34.2) | 42 (35.0) | 9 (7.5) | 8 (6.7) |
| | *p* value | | **0.047** | 0.059 | 0.109 | 0.948 |
| **Major EPS composition** | Polysaccharides | 13 (10.8) | 12 (10.0) | 11 (9.2) | 5 (4.2) | 1 (0.8) |
| | Proteins | 36 (30.0) | 35 (29.2) | 28 (23.3) | 8 (6.7) | 6 (5.0) |
| | DNA | 39 (32.5) | 38 (31.2) | 38 (31.2) | 11 (9.2) | 9 (7.5) |
| | Unknown | 32 (26.7) | 29 (24.2) | 31 (25.8) | 12 (10.0) | 4 (3.3) |
| | *p* value | | 0.503 | **0.015** | 0.494 | 0.511 |

The classification of biofilm production, EPS composition, and the presence of biofilm-associated genes is shown for all isolates (*n* = 120). Bold text denotes statistically significant *p* values.

38.3% (46/120). The major EPS components were DNA (32.5%), proteins (30.0%), and poly-saccharides (10.8%); the remaining 27.7% could not be identified (Table 1).

### Detection of biofilm-associated genes

Genes *fsnR*, *rmlA*, *rpfF-2*, and *xanB* were detected in 95.0% (114/120), 90.0% (108/120), 30.0% (36/120), and 16.7% (20/120) of the isolates, respectively (Table 1). However, only the presence of the *fsnR* gene was associated with biofilm production ($p$ = 0.047). The frequency of $rmlA^+$/$rpfF$-$2^-$ strains was high (n = 72; 60%), indicating that the presence of the *rmlA* gene was inversely associated to the *rpfF-2* variant gene (S1 Table). In addition, the presence of the *rmlA* gene was associated with the frequency of polysaccharides, proteins, or extracellular DNA as a major EPS component ($p$ = 0.015).

### Potential biofilm biomarker peaks

The analysis of gene-expressing groups with different biofilm-producing stages led to three possible biomarker peaks: 4661 *m/z*, 6074 *m/z*, and 6102 *m/z*. The absence or presence of these peaks was associated with specific *rmlA* and *rpfF-2* genotypes (Fig 1). The $rmlA^+$ genotype was associated with the presence of the 4661 *m/z* peak ($p$ = 0.000; 94.5% sensitivity, 17.0% specificity, 63.9% PPV, and 66.7% NPV) and the absence of the 6074 *m/z* peak, along with the presence of the 6102 *m/z* peak ($6074^-$/$6102^+$; $p$ = 0.000; Fig 2), which had a 94.3% sensitivity, 40.0% specificity, 91.7% PPV, and 50.0% NPV. The $rpF$-$2^-$ genotype was associated with the presence of the 4661 *m/z* peak ($p$ = 0.000; 93.6% sensitivity, 45.2% specificity, 52.4% PPV, and 91.7% NPV) and the $6074^+$/$6102^-$ phenotype ($p$ = 0.007; 65.7% sensitivity, 0.0% specificity, 82.1% PPV and 0.0% NPV; Fig 3). All three potential biomarker peaks were tentatively assigned to a 50S ribosomal protein, with MASCOT score of 27─32 and protein sequence coverage of 79%─98% (L33 for 4661 and 6102 *m/z*, and L30 for 6074 *m/z*).

## Discussion

All *S. maltophilia* isolates were able to form biofilm and the majority were either moderate or strong producers. Biofilm formation is a virulence factor that significantly contributes to the bacteria's progression of disease in the lungs of patients with cystic fibrosis [2]. Although none of our patients had cystic fibrosis, *S. maltophilia* strains showed high biofilm production. Indeed, non-cystic fibrosis strains produce biofilm more efficiently than cystic fibrosis strains [17]. Therefore, the possibility that biofilm formation contributes to the severity of the infection cannot be discarded. Furthermore, the capability to form biofilm might aid the dissemination of the pathogen in the hospital setting [17].

 *S. maltophilia* EPS composition was highly heterogeneous; a similar proportion of proteins, extracellular DNA, and polysaccharides was observed. The specific composition of the EPS influences the successful protection against antimicrobial agents. Some proteins are preferentially expressed in biofilm cells compared with planktonic cells and participate in adhesion, biofilm stability and antibiotic resistance [7]. The cation-chelating properties of extracellular DNA disrupt the bacterial cell membrane which affects positively charged antibiotics [6, 7]. Negatively charged β-linked polysaccharides (e.g., cellulose and chitin) are fundamental components of *S. maltophilia* EPS [18, 19] and bind to positively charged antibiotics effectively inhibiting their antimicrobial effect.

 The generation and maintenance of the biofilm matrix depends on the regulation of cell motility and flagella synthesis. We detected a high frequency of the *fsnR* gene, which encodes a flagellum biosynthesis regulator that activates the transcription of flagellar genes; this high gene frequency could also be related to the high biofilm production in our strain population.

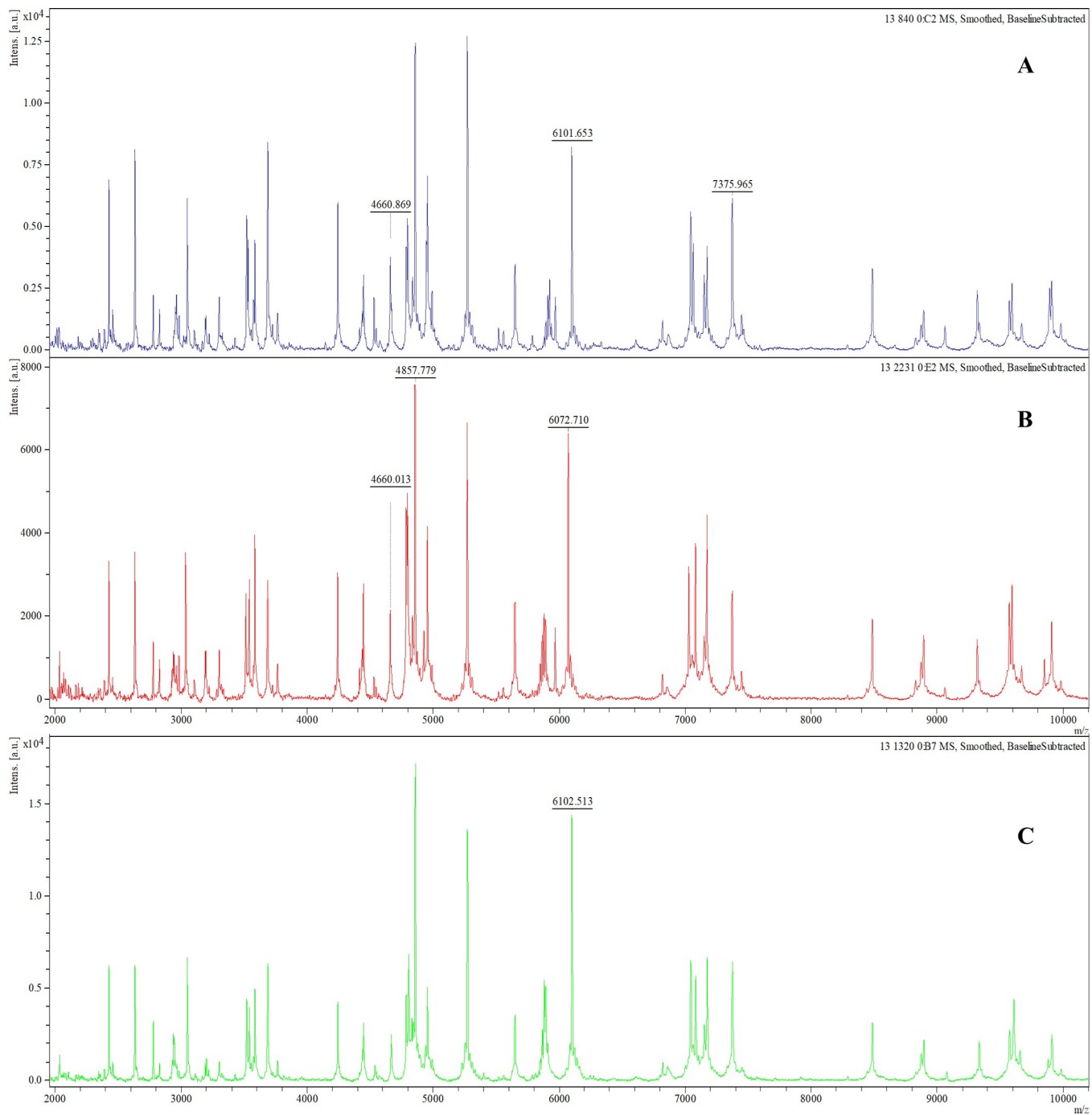

**Fig 1. Representative mass spectra comparison of potential biomarker peaks of *rmlA⁺* and *rpF-2⁻* genotype in biofilm-producing *S. maltophilia* isolates.** In the *rmlA⁺* genotype the 4661 *m/z* and 6102 *m/z* peaks are present and the 6074 *m/z* peak is absent (**A**) whereas in the *rmlA⁻* genotype the 4661 *m/z* and 6074 *m/z* peaks are present and the 6102 *m/z* peak is absent (**B**). In the *rpF-2⁻* genotype the 4661 *m/z* and 6074 *m/z* peaks are absent and the 6102 *m/z* peak is present.

The *rmlA* gene encodes a glucose-1-phosphate thymidyl transferase and *xanB* gene encodes a phosphomannoseisomerase/GDP-mannosepyrophosphorylase, both of which are necessary for the lipopolysaccharide (LPS) O-antigen biosynthesis, which is required for biofilm

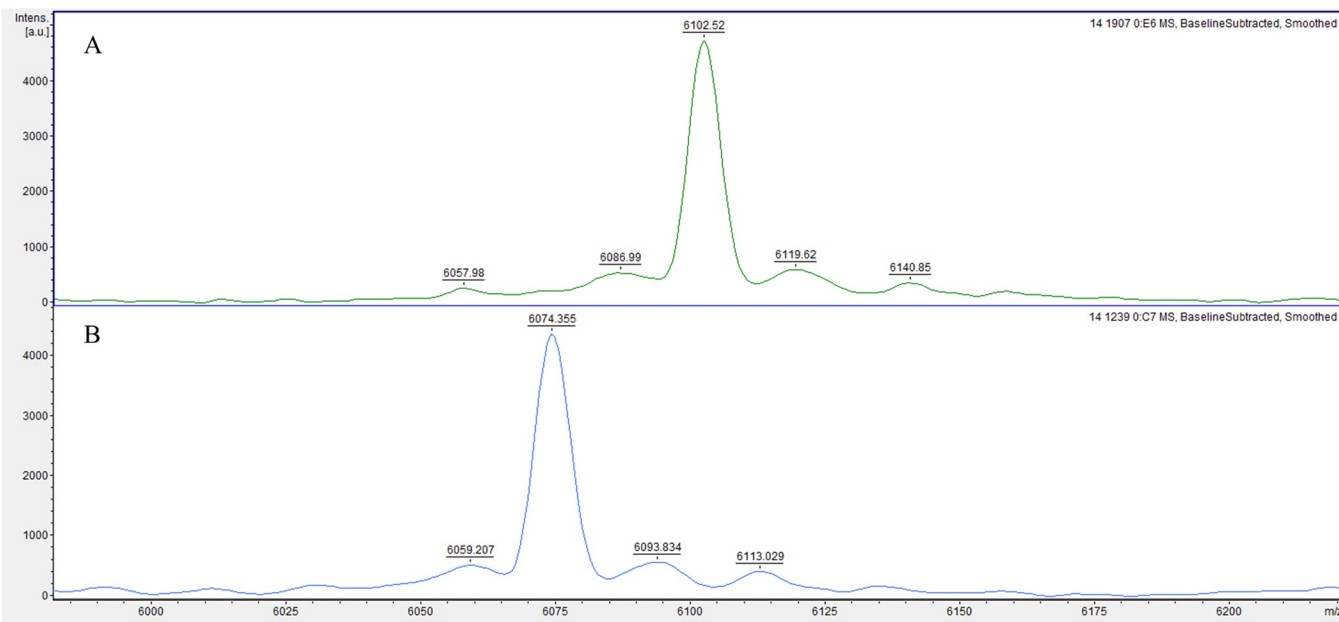

**Fig 2. Representative mass spectra comparison of two potential biomarker peaks of *rmlA* genotype in biofilm-producing *S. maltophilia* isolates.** The *rmlA*+ genotype group exhibits a 6074⁻/6102⁺ phenotype (**A**), whereas the *rmlA*⁻ genotype group exhibits a 6074⁺/6102⁻ phenotype (**B**).

formation and twitching motility [15]. The *rmlA* gene exhibits a high prevalence in *S. malto-philia* strains (65.2–97.7%) [5], similar to our results. In strains where polysaccharides, proteins, or extracellular DNA were a major EPS component, the *rmlA* gene was present.

The *rpf* (regulation of pathogenicity factors) gene cluster contains an enoyl coenzyme A hydratase (RpfF), which acts as a diffusible signal factor (DSF) synthase. DSF quorum sensing regulates swimming and twitching motilities, biofilm biomass and microcolony formation, and the development of biofilm's structure [5, 15]. In *S. maltophilia*, the *rpf* cluster possesses two variants that differentially regulate DSF production and detection. *rpf-1* strains produce detectable DSF whereas *rpf-2* strains are DSF-deficient and have to detect DSF produced by neighbouring bacteria [20, 21]. The frequency of both *rpfF* gene variants differs; *rpf-1* strains can be up to 55.5% and the *rpfF-2* variant, which has been mistakenly reported as *rpfF*-negative strains [14, 22], are actually present in 44.5–83.5% of the strains [5, 20, 21]. Almost a third of our population was *rpfF-2* positive. Further studies should include the search of both *rpfF* gene variants. The frequency of the *rmlA*⁺/*rpfF*⁻ genotype in our strains was high, which has previously associated with non-cystic fibrosis strains (as is our case) and strong biofilm production [22].

Other genes associated with biofilm production in *S. maltophilia* include those related to LPS production (*spgM*, *rmlC and xanA*), fimbriae (*smf-1*), flagellar biosynthesis (*fliA*), quorum sensing (*ax21*), biofilm and swimming motility regulator (*bsmR*), and the gene for glycolytic enzyme phosphoglycerate mutase (*gpmA*) [2, 14, 15, 23, 24]. Further studies of these specific genes in our strain population should also be conducted to better characterize the genetic mechanisms of biofilm production regulation.

MALDI-TOF MS can efficiently identify bacteria directly in samples prepared from biofilms [8] and discriminate between biofilm-producer and non-producer strains, using the same method and software that is used in the identification stage [25]. Among the three potential biomarker peaks of biofilm production we detected, the 6074⁻/6102⁺ phenotype was associated with the *rmlA*⁺ genotype, which could be used as a potential biomarker peak of strong

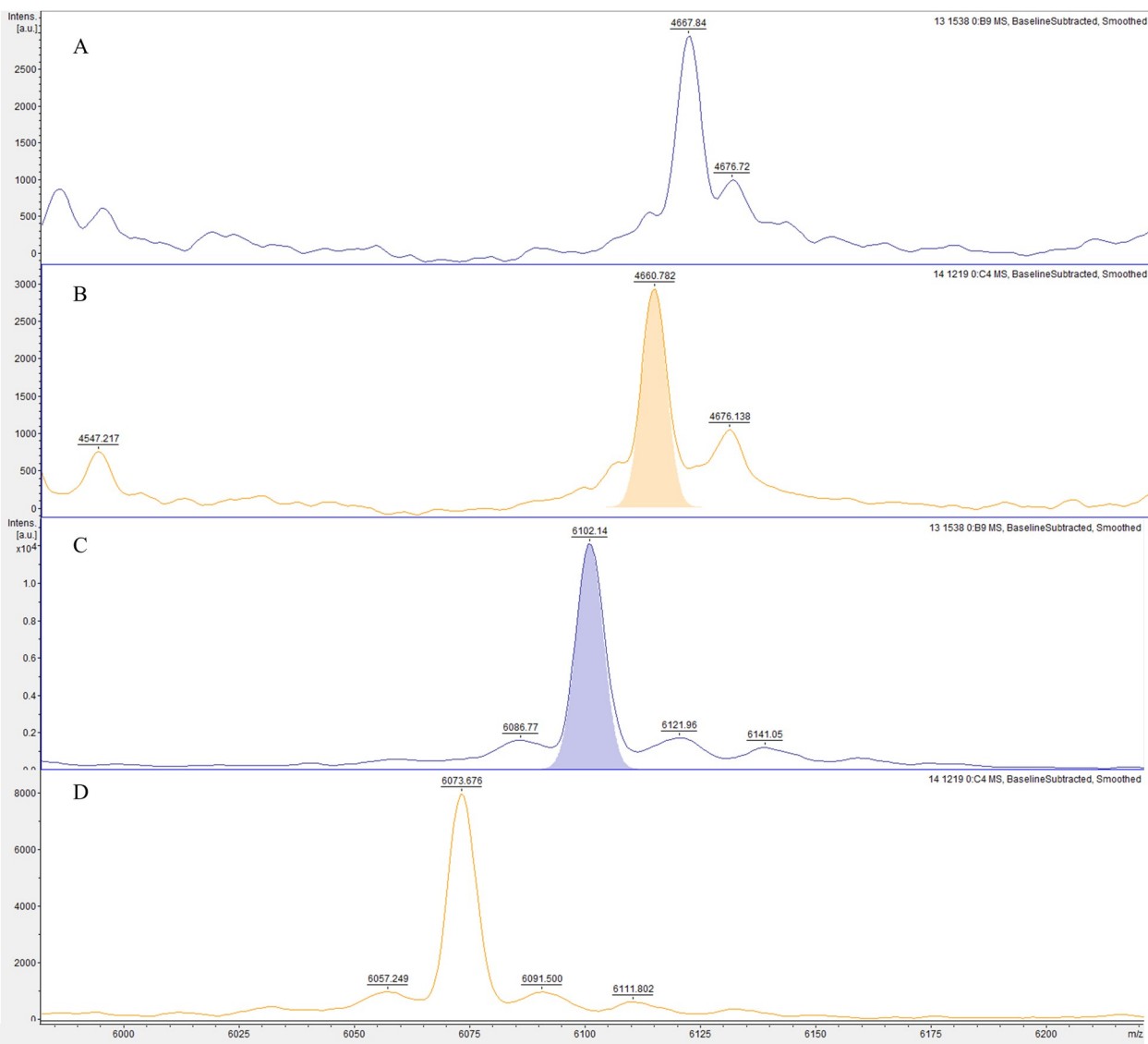

**Fig 3. Representative mass spectra comparison of two potential biomarker peaks of *rpfF* genotype in biofilm-producing *S. maltophilia* isolates.** The *rpfF*+ genotype group exhibits a 4661- (**A**) and a 6074-/6102+ (**C**) phenotype, whereas the *rpfF*- genotype group exhibits a 4661+ (**B**) and a 6074+/6102- (**D**) phenotype.

biofilm production in *S. maltophilia*. MALDI-TOF MS analyses showed a correlation between the relative abundance of 50S ribosomal proteins (L30 and L33) and the presence of the *fnsR*, *rmlA* and *rpfF*-2 genes, which have been suggested earlier to play a role in biofilm formation. Further studies could allow the identification of novel protein candidates involved in biofilm-matrix formation.

ICU-associated *S. maltophilia* infections seem to behave differently than those associated to the community [12]. *S. maltophilia* hospital-acquired pneumonia occurs in severe, long-stay intensive care patients with prolonged invasive ventilation, and in-hospital mortality can be as high as 49.7% [26]. ICU-obtained *S. maltophilia* isolates showed low clonality, 12 different sequence types and two isolates with the same allelic profile, corresponding to ST186. *S. maltophilia* strains from the same hospital show low clonal relatedness [9, 10, 17, 20, 27, 28] and a

high degree of genetic diversity [20, 21, 27, 29], suggesting that person-to-person is not the major mode of transmission [28]. The transmission of *S. maltophilia* within the hospital setting is rarely seen, and it usually occurs as a result of a cross-contamination issue [30]. Nevertheless, contained *S. maltophilia* outbreaks within the ICU of a hospital setting have been reported associated with contaminated bronchoscopes [12], and superficial surfaces near the patient, such as shower outlets [31], faucets [32] and drinking water supplying devices for ICU patients [33].

The identification of sources of *S. maltophilia* infections might help to implement control measures, such as the comparison of clinical and environmental strains, as the former are more likely to produce biofilm than the latter [34]. However, the high genetic diversity exhibited by *S. maltophilia* strains restricts its use in epidemiologic studies [35]; ST distribution does not account for geographical or clinical origin [29]. In fact, some recent studies still report new ST [20, 21, 27, 36], differing in virulence and resistance genes [36], reflecting the high genetic variability rate of this pathogen.

A limitation of the present study is that we did not include the analysis of mass spectra of bacteria on biofilm-state, which could have aided greatly in our comparison. Additionally, non-biofilm-producing isolates should be included in the comparison studies, but we did not find any. Nevertheless, current limitations when characterizing biofilm-producing strains include the biofilm production variability, which depends on EPS specific composition, and infection type, stage, and location [5].

## Conclusion

The characterization of biofilm production, EPS composition, associated genes and mass spectra in *S. maltophilia* clinical isolates yielded three potential biomarker peaks for the *rmlA*$^+$ and *rpfF-2*$^-$ genotype by MALDI-TOF MS, which could potentially be used to discriminate biofilm production in *S. maltophilia*. ICU-obtained *S. maltophilia* isolates showed high genetic diversity, suggesting no transmission within the hospital ward. Molecular approaches such as the detection of biofilm-associated genes could be used to promptly recognize biofilm-producing *S. maltophilia* strains or specific EPS composition and aid in the establishing of an appropriate antibiotic therapy.

## Supporting information

**S1 Fig. Genetic diversity of ICU-obtained *S. maltophilia* clinical isolates.** Fifty *S. maltophilia* isolates obtained from the ICU from both hospitals were analyzed by PFGE and MLST. ICU: Intensive care unit; ND: Not determined.
(TIF)

**S1 Table. Clinical and demographic characteristics of patients included in the study.** The results obtained regarding biofilm production, EPS major composition, presence of biofilm-associated genes, and potential biofilm biomarker peaks are also included.
(XLSX)

## Author Contributions

**Conceptualization:** Elvira Garza-González, Samantha Flores-Treviño.

**Data curation:** Óscar Manuel Alonso-Ambriz, Gabriela Aimee Salazar-Mata.

**Formal analysis:** Paola Bocanegra-Ibarias, Samantha Flores-Treviño.

**Funding acquisition:** Samantha Flores-Treviño.

**Investigation:** Edeer Montoya-Hinojosa, Paola Bocanegra-Ibarias, Eduardo Pérez-Alba.

**Methodology:** Edeer Montoya-Hinojosa, Paola Bocanegra-Ibarias, Óscar Manuel Alonso-Ambriz, Gabriela Aimee Salazar-Mata, Samantha Flores-Treviño.

**Project administration:** Elvira Garza-González, Adrián Camacho-Ortiz, Samantha Flores-Treviño.

**Resources:** Elvira Garza-González, Licet Villarreal-Treviño, Eduardo Pérez-Alba, Rayo Morfín-Otero, Eduardo Rodríguez-Noriega.

**Supervision:** Paola Bocanegra-Ibarias, Elvira Garza-González, Licet Villarreal-Treviño, Adrián Camacho-Ortiz, Samantha Flores-Treviño.

**Validation:** Samantha Flores-Treviño.

**Visualization:** Eduardo Pérez-Alba, Samantha Flores-Treviño.

**Writing – original draft:** Edeer Montoya-Hinojosa, Samantha Flores-Treviño.

**Writing – review & editing:** Elvira Garza-González, Adrián Camacho-Ortiz, Samantha Flores-Treviño.

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
