## [Decision Letter · Decision Letter 0]

11 Nov 2020

PONE-D-20-31198

Discrimination of biofilm-producing Stenotrophomonas maltophilia clinical strains by matrix-assisted laser desorption ionization–time of flight

PLOS ONE

Dear Dr. Flores-Treviño,

Thank you for submitting your manuscript to PLOS ONE. After careful consideration, we feel that it has merit but does not fully meet PLOS ONE’s publication criteria as it currently stands. Therefore, we invite you to submit a revised version of the manuscript that addresses the points raised during the review process.

I have chosen to send you constructive comments rather than waiting for the second referee comments. In any case, the editor omments would have been the same. 

Your manuscript presents an excellent mix of structural characterization by MALDI-TOF-MS of the biofilm produced by the Gram-negative bacillus *S. maltophilia with* molecular approaches such as screening  for the presence of film-associated genes, in which the fsnR gene was associated with biofilm production. And you have shown that the rmlA + genotype was associated with the rpfF – genotype.

The molecular approaches you have used consisted of biofilm production and EPS composition determination, biofilm-associated genes detection, biofilm-associated genes detection, clonal diversity determination, multilocus sequence typing, and identity assignment and statistical analyses

The specific aim of this manuscript was to search for MALDI-TOF-MS protonated 50S ribosomal protein biomarker molecules that could rapidly detect biofilm production in *S. maltophilia *clinical isolates obtained from care hospitals in Mexico. You have indicated that the components of this biofilm matrix was a mixture of polysaccharides, proteins, nucleic acids, and lipids that varies among bacterial species.

According to your results MALDI-TOF-MS can be used as a comparison tool to identify  yielded three protonated 50S ribosomal protein biomarker molecules at m/z 4661, 6074, and 6102 m/z obtained from the  biofilm-producing rmlA + and rpfF - genotypes with >90% sensitivity ( p <0.001).  In addition, you indicate that the MALDI-TOF-MS analyses showed a correlation between the relative abundance of 50S ribosomal proteins (L30 and L33) and the presence of the fnsR, rmlA and rpfF -2 genes.

My main concern with your manuscript is the complete absence of any real MALDI-TOF-MS spectra !

What you are showing in your Fig 1. Representative mass spectra comparison of two potential biomarker peaks of *relax *genotype in biofilm-producing *S. maltophilia *isolates. And Fig 2. Representative mass spectra comparison of two potential biomarker peaks of *rpfF *genotype in biofilm-producing *S. maltophilia *isolates.

These figures 1 and 2 are not complete MALDI-TOF-MS spectra! They are zooms enhancing the diagnostic ions.  And there is no way to look at the complete comparison portraits. Considering that the whole work presented in your manuscript depends on the MALDI-TOF-MS detections of three protonated 50S ribosomal protein biomarker molecules at m/z 4661, 6074, and 6102*.*

 I therefore encourage you to include the complete MALDI-TOF-MS and you can indicate with a different colors your three biomarkers at *m/z* 4661, 6074 and 6102.  Having done that, you can incorporate you actual Figs 1 and 2.

.

that your decision is justified on PLOS ONE’s publication criteria and not, for example, on novelty or perceived impact.

We look forward to receiving your revised manuscript.

Kind regards,

Joseph Banoub, Ph,D., D. Sc.

Academic Editor

PLOS ONE

Journal Requirements:

2. Please note that PLOS does not permit references to “data not shown.” Authors should provide the relevant data within the manuscript, the Supporting Information files, or in a public repository. If the data are not a core part of the research study being presented, we ask that authors remove any references to these data.

3.In your Methods section, please provide additional information about the medical data/samples collected and the demographic details of the human subjects. Please ensure you have provided sufficient details to replicated the analyses such as: a) a description of any inclusion/exclusion criteria that were applied to the medical records/samples, b) a table of relevant demographic details.

4. To comply with PLOS ONE submission guidelines, in your Methods section, please provide additional information regarding your statistical analyses. For more information on PLOS ONE's expectations for statistical reporting, please see https://journals.plos.org/plosone/s/submission-guidelines.#loc-statistical-reporting.

EDITOR COMMENTS

I have chosen to send you constructive comments rather than waiting for the second referee comments. In any case, the editor omments would have been the same.

Your manuscript presents an excellent mix of structural characterization by MALDI-TOF-MS of the biofilm produced by the Gram-negative bacillus S. maltophilia with molecular approaches such as screening for the presence of film-associated genes, in which the fsnR gene was associated with biofilm production. And you have shown that the rmlA + genotype was associated with the rpfF – genotype.

The molecular approaches you have used consisted of biofilm production and EPS composition determination, biofilm-associated genes detection, biofilm-associated genes detection, clonal diversity determination, multilocus sequence typing, and identity assignment and statistical analyses

The specific aim of this manuscript was to search for MALDI-TOF-MS protonated 50S ribosomal protein biomarker molecules that could rapidly detect biofilm production in S. maltophilia clinical isolates obtained from care hospitals in Mexico. You have indicated that the components of this biofilm matrix was a mixture of polysaccharides, proteins, nucleic acids, and lipids that varies among bacterial species.

According to your results MALDI-TOF-MS can be used as a comparison tool to identify yielded three protonated 50S ribosomal protein biomarker molecules at m/z 4661, 6074, and 6102 m/z obtained from the biofilm-producing rmlA + and rpfF - genotypes with >90% sensitivity ( p <0.001). In addition, you indicate that the MALDI-TOF-MS analyses showed a correlation between the relative abundance of 50S ribosomal proteins (L30 and L33) and the presence of the fnsR, rmlA and rpfF -2 genes.

My main concern with your manuscript is the complete absence of any real MALDI-TOF-MS spectra !

What you are showing in your Fig 1. Representative mass spectra comparison of two potential biomarker peaks of relax genotype in biofilm-producing S. maltophilia isolates. And Fig 2. Representative mass spectra comparison of two potential biomarker peaks of rpfF genotype in biofilm-producing S. maltophilia isolates.

These figures 1 and 2 are not complete MALDI-TOF-MS spectra! They are zooms enhancing the diagnostic ions. And there is no way to look at the complete comparison portraits. Considering that the whole work presented in your manuscript depends on the MALDI-TOF-MS detections of three protonated 50S ribosomal protein biomarker molecules at m/z 4661, 6074, and 6102.

I therefore encourage you to include the complete MALDI-TOF-MS and you can indicate with a different colors your three biomarkers at m/z 4661, 6074 and 6102. Having done that, you can incorporate you actual Figs 1 and 2.

.

Reviewers' comments:

Reviewer's Responses to Questions

**Comments to the Author**

1. Is the manuscript technically sound, and do the data support the conclusions?

Reviewer #1: Yes

2. Has the statistical analysis been performed appropriately and rigorously? 

Reviewer #1: Yes

3. Have the authors made all data underlying the findings in their manuscript fully available?

Reviewer #1: Yes

4. Is the manuscript presented in an intelligible fashion and written in standard English?

Reviewer #1: Yes

5. Review Comments to the Author

Reviewer #1: The authors collected a total of 120 S. maltophilia clinical isolates and evaluated their biofilm-forming ability, screened the presence of biofilm-associated genes, and employed MALDI-TOF to identify potential biofilm-associated peaks. The topic is of general interest and the findings will provide a significant contribution to clinical diagnosis. Below are my comments on this manuscript

comments:

1. Line 161 and Line 356, UCI-obtained? A typo?

2. Line 176, Please include a reference for BLAST program.

3. Line 214, the supplementary Figure is not provided.

4. In Table 1, please state why does the number in the parentheses stand for?

5. In Table 1, please use rpfF-2 for consistency.

6. In Table 1, does the term "Other" mean that an unknown component other than polysaccharides, proteins, and DNA play a major role in the biofilm?

6. PLOS authors have the option to publish the peer review history of their article (what does this mean?). If published, this will include your full peer review and any attached files.

Reviewer #1: No

---

## [Author Response · Author response to Decision Letter 0]

1 Dec 2020

Reviewer #1 Comments to the Author

1. Line 161 and Line 356, UCI-obtained? A typo?

Response: Yes, it is indeed a typo. We apologize for the inconvenience. We have corrected the typo from UCI to ICU.

2. Line 176, Please include a reference for BLAST program.

Response: We have added the following reference: Altschul SF, Gish W, Miller W, Myers EW, Lipman DJ. Basic local alignment search tool. J Mol Biol. 1990;215(3):403-10. doi: 10.1016/S0022-2836(05)80360-2. PubMed PMID: 2231712.

3. Line 214, the supplementary Figure is not provided.

Response: We apologize for the inconvenience. The supplementary Figure S1 Fig has been added to the Supplementary files. 

4. In Table 1, please state why does the number in the parentheses stand for? 

Response: The number inside the parenthesis stands for the percentage out of the total analyzed strains, which is 120. This total number is included in the text and in the Table title. 

5. In Table 1, please use rpfF-2 for consistency.

Response: We have made corrected rpf to rpf-2. 

6. In Table 1, does the term "Other" mean that an unknown component other than polysaccharides, proteins, and DNA play a major role in the biofilm?

Response: Yes, “Other” refers to EPS components different from polysaccharides, proteins, and DNA. We have changed the term to “Unknown” to avoid confusion.

---

## [Editor Report · Decision Letter 1]

16 Dec 2020

Discrimination of biofilm-producing Stenotrophomonas maltophilia clinical strains by matrix-assisted laser desorption ionization–time of flight

PONE-D-20-31198R1

Dear Dr. Flores-Treviño,

We’re pleased to inform you that your manuscript has been judged scientifically suitable for publication and will be formally accepted for publication once it meets all outstanding technical requirements.

Kind regards,

Joseph Banoub, Ph,D., D. Sc., FRSC

Academic Editor

PLOS ONE
---

## [Editor Report · Acceptance letter]

18 Dec 2020

PONE-D-20-31198R1 

Discrimination of biofilm-producing *Stenotrophomonas maltophilia* clinical strains by matrix-assisted laser desorption ionization–time of flight 

Dear Dr. Flores-Treviño:

I'm pleased to inform you that your manuscript has been deemed suitable for publication in PLOS ONE. Congratulations! Your manuscript is now with our production department. 

Kind regards, 

on behalf of

Dr. Joseph Banoub 

Academic Editor

PLOS ONE